# Effect of Chemical Chaperones on the Stability of Proteins during Heat– or Freeze–Thaw Stress

**DOI:** 10.3390/ijms241210298

**Published:** 2023-06-18

**Authors:** Vera A. Borzova, Tatiana B. Eronina, Valeriya V. Mikhaylova, Svetlana G. Roman, Andrey M. Chernikov, Natalia A. Chebotareva

**Affiliations:** Bach Institute of Biochemistry, Federal Research Centre “Fundamentals of Biotechnology” of the Russian Academy of Sciences, Leninsky pr. 33, 119071 Moscow, Russia; vera.a.borzova@gmail.com (V.A.B.); eronina@inbi.ras.ru (T.B.E.); mikhaylova.inbi@inbox.ru (V.V.M.); svetabaj@gmail.com (S.G.R.); chernikov.andrei.m@gmail.com (A.M.C.)

**Keywords:** glutamate dehydrogenase, glycogen phosphorylase *b*, protein aggregation, protein stability, chemical chaperones, osmolytes, 2-hydroxypropyl-β-cyclodextrin, freeze–thaw stress

## Abstract

The importance of studying the structural stability of proteins is determined by the structure–function relationship. Protein stability is influenced by many factors among which are freeze–thaw and thermal stresses. The effect of trehalose, betaine, sorbitol and 2-hydroxypropyl-β-cyclodextrin (HPCD) on the stability and aggregation of bovine liver glutamate dehydrogenase (GDH) upon heating at 50 °C or freeze–thawing was studied by dynamic light scattering, differential scanning calorimetry, analytical ultracentrifugation and circular dichroism spectroscopy. A freeze–thaw cycle resulted in the complete loss of the secondary and tertiary structure, and aggregation of GDH. All the cosolutes suppressed freeze–thaw- and heat-induced aggregation of GDH and increased the protein thermal stability. The effective concentrations of the cosolutes during freeze–thawing were lower than during heating. Sorbitol exhibited the highest anti-aggregation activity under freeze–thaw stress, whereas the most effective agents stabilizing the tertiary structure of GDH were HPCD and betaine. HPCD and trehalose were the most effective agents suppressing GDH thermal aggregation. All the chemical chaperones stabilized various soluble oligomeric forms of GDH against both types of stress. The data on GDH were compared with the effects of the same cosolutes on glycogen phosphorylase *b* during thermal and freeze–thaw-induced aggregation. This research can find further application in biotechnology and pharmaceutics.

## 1. Introduction

Protein stability attracts such a close attention of scientists because of the undeniable relation between the correct functioning of proteins and their structure, and is of great interest for the biotechnology, pharmaceutical and food industries. An understanding of protein stability is essential for optimizing the expression, purification, formulation, storage and structural studies of proteins [1]. The impact of various negative factors on the structure of proteins can lead to their dysfunction, while the existence of in vivo systems of cellular proteostasis ensures the stability of the proteome and, thus, the resistance of cells to various stresses. A variety of components of the cellular proteostasis system, such as a pool of chaperone proteins, mechanisms of protein disaggregation and refolding, accumulation of osmolytes under stress, apparently evolved to help the cell cope with the formation of non-functional harmful aggregates by all available means. Nowadays, both the mechanism of action and the contribution of each component of the cellular proteostasis system to the proteome stability remain the subject of scientific investigations. Protein stability, i.e., the ability of proteins to maintain their structural organization and/or biological activity, is very sensitive to environmental conditions and the presence of various agents. In particular, a change in the temperature regime can lead to the loss of stability of the protein molecule, its denaturation and subsequent aggregation.

The effects of high and low temperatures on proteins are different. Heat denaturation of proteins is entropically driven, as it is connected with the increase in conformational entropy due to protein unfolding [2]. Protein freezing is widely used in pharmaceuticals in manufacturing [3] and during long-term storage of proteins [4]. However, the process of freezing and subsequent thawing can lead to conformational changes in the protein molecule and its cold denaturation [5,6]. This type of denaturation is enthalpically driven [5,7]. Water plays a key role in cold denaturation and repulsive interactions between it and non-polar protein residues become weaker, resulting in partial unfolding of the protein [8]. Low temperatures are not the only factor influencing the change in protein structure. The most important destabilizing factor is the formation of ice [9], because the direct interaction of the protein with the ice surface causes a perturbation that changes the native protein folding [9]. Strambini and Gabelleri [9] supposed that some additives were able to cover the ice surface, thereby reducing the proteins adsorption on it, and that the preferential exclusion of the cosolutes from the protein, which stabilized their native folding, might increase the stability of the protein at the ice-water interface. Further, the presence of cosolutes can affect the protein hydration layer, which affects protein structure, thermodynamics, stability, and activity [10].

Osmolytes, which tend to accumulate in the cell in high concentrations in response to stress [11,12], can act as such cosolutes. Stabilizing, or compatible, osmolytes are low-molecular compounds of various nature. The common feature of them is the ability to shift the structure of proteins towards the folded state without disturbing their structure and function. It is generally accepted that compatible osmolytes are preferentially excluded from the protein surface due to thermodynamic aversion to the protein backbone, and their main stabilizing effect is due to preferential hydration of the protein surface [13]. A stable hydration shell acts as a barrier that protects proteins from denaturation and aggregation and preserves the functional activity of proteins. This principle of protein stabilization has been shown for osmolytes of various classes, for example, trehalose (Tre, sugar) [14,15,16], betaine (Bet, methylamine) [17,18], sorbitol (Sorb, polyol) [19,20]. Due to the osmolytes ability to stabilize proteins, prevent their aggregation, and increase their functional activity under different kinds of stress and variable conditions, these compounds are often called “chemical chaperones” [21,22,23].

In addition to osmolytes, the role of protective cosolutes can be performed by cyclodextrins, called “artificial chaperones”, or their derivatives, in particular 2-hydroxypropyl-β-cyclodextrin (HPCD). HPCD is an oligosaccharide and has properties similar to surfactants [24], thus it can play a dual role in stabilizing proteins acting as sugar-like chemical chaperones and as surfactant. HPCD has high water solubility and low toxicity. Numerous works have been published regarding the protective effect of HPCD on thermal and chemical denaturation of proteins, explaining that HPCD forms complexes with accessible hydrophobic amino acid residues [24,25,26,27]. HPCD can inhibit protein aggregation at air-water, ice-water interfaces during freeze–thawing, exhibiting similarities with surfactants action [24]. In some cases, HPCD can provoke protein destabilization and accelerate their thermal denaturation and aggregation [28,29,30].

A lot of scientific works have been devoted to the study of the heat stress effect on proteins, as well as methods of protection against it. As for cold stress, its effect on protein stability has been studied to a lesser extent, despite the obvious relevance of such studies. The role of various cosolutes in the cold stress conditions also remains poorly understood. Therefore, the aim of this work was to study the effect of osmolytes (Tre, Bet, Sorb) and HPCD on protein stability during freezing–thawing and to compare it with the effect of the cosolutes on protein stability under heat stress.

Glutamate dehydrogenase from bovine liver (GDH) was chosen as a model protein. GDH is a homohexamer with six identical subunits with a molecular mass of 56–57 kDa located in two trimers [31]. The mechanism of thermal denaturation and aggregation of GDH at 50 °C has been studied in detail in our previous work [32]. The behavior of GDH during freezing–thawing and the effect of osmolytes or HPCD on this process remain unexplored at the moment.

Another oligomeric protein with well-studied thermal aggregation mechanism was used as a comparison object. Glycogen phosphorylase *b* from rabbit skeletal muscle (Ph*b*) is a dimeric protein formed by two identical subunits with a molecular weight of 97.4 kDa [33]. Thermal aggregation of Ph*b* at 48 °C [34] and the effects of Tre, Bet [35,36,37,38], and HPCD [29] on it were previously studied in detail by our team. It was shown that Tre and Bet effectively protected Ph*b* from thermal denaturation and aggregation. HPCD destabilized Ph*b* and accelerated its thermal aggregation. However, there are no data on the stability of Ph*b* during freezing–thawing and the effect of osmolytes or HPCD on this process.

The aim of this work was to compare the effect of chemical chaperones on the stability of two model proteins (GDH and Ph*b*) during freeze–thaw and thermal stress using several methods: the dynamic light scattering (DLS), differential scanning calorimetry (DSC), analytical ultracentrifugation (AUC) and circular dichroism (CD) spectroscopy.

## 2. Results

### 2.1. Effect of the Chemical Chaperones on the Kinetic Stability of GDH under the Thermal Stress or after a Freeze–Thaw Cycle

The kinetic curves of GDH aggregation at 50 °C in the absence or presence of the chemical chaperones were registered by measuring the light scattering intensity (*I*) increase over the time (*t*) using DLS. The examples of these kinetic curves with different chemical chaperones are presented in Figure 1A.

The *I*(*t*) dependences were obtained at several chaperone concentrations for each studied compound. These dependences were approximated by Equation (1) and the parameters *v*_0_ and *v*_0,add_ (see Section 4) were calculated. The dependences *v*_0,add_/*v*_0_ on the concentration of the chemical chaperones, [L], are shown in Figure 1B. These data were analyzed using Equation (2), and the parameter [L]_0.5_, characterizing the anti-aggregation activity of a chemical chaperone, was calculated for each chemical chaperone (Table 1). Tre and HPCD appear to be the most effective chemical chaperones in the case of GDH thermal aggregation.

Preliminary experiments have shown that frozen–thawed GDH (GDH_fr_) tends to aggregate and precipitate at room temperature; therefore, the aggregation of GDH_fr_ was studied at 25 °C. The kinetic curves of GDH_fr_ aggregation in the absence or presence of different chemical chaperones are presented in Figure 1C. The *I*(*t*) dependences were also obtained at several concentrations of each of the studied compounds, and the initial aggregation rates were calculated using Equation (1). The dependences of (*v*_0,add_/*v*_0_) on the chemical chaperones concentration are shown in Figure 1D. The parameter [L]_0.5_ was calculated for Tre, Sorb and Bet. Among these chemical chaperones Bet is the most effective (Table 1), with [L]_0.5_ = 0.53 ± 0.02 mM, that is two orders of magnitude less than for Tre or Sorb. In the case of HPCD we observed a striking effect of aggregation suppression even at micromolar concentrations of HPCD (see the inset in Figure 1D). The calculation of the [L]_0.5_ value in this range of concentrations turned out to be impossible.

To investigate these particular effects of HPCD and Bet we calculated the average hydrodynamic radii (*R*_h_) of protein aggregates in the presence of low concentrations of HPCD or Bet (1 mM) as well as their relatively high concentrations (50 mM HPCD and 30 mM Bet), which fully suppressed GDH_fr_ aggregation kinetics. The dependences of *R*_h_ on time are shown in Figure 2A,C.

The *R*_h_ values split into two populations in the absence or presence of the chemical chaperones. One of the populations (1) shown in Figure 2A,C by open circles consists of large particles which *R*_h_ can reach more than 1 μm, while the second one (2) has a significantly lower *R*_h_ value in the range from tens to 200–250 nm. Notably, in the presence of 1 mM HPCD the large-sized aggregates population shows no growth of *R*_h_ over the time. When analyzing the DLS data one should remember that this method is focused on larger particles which make the major contribution to the light scattering intensity. Thus, the kinetic curves of *I*(*t*) mainly reflect the behavior of these large-sized particles, which explains the apparent absence of the aggregation kinetics at low HPCD concentration. Therefore, the [L]_0.5_ values estimated from these *I*(*t*) dependences are so small (Table 1). A relatively high concentration of HPCD (50 mM) results in the formation of smaller particles in both (1) and (2) aggregate populations; these populations become comparable in size. The similar data were obtained for Bet (Figure 2C), where 1 mM of the chemical chaperone demonstrated the growth of population (1), while 30 mM Bet suppressed the growth but did not prevent the formation of large aggregates. The calculated particle size on number distributions (Figure 2B,D) demonstrates the quantitative predominance of small-sized aggregate populations in all cases. It can be seen from Figure 2B,D that the increase in the chemical chaperones concentration leads to the decrease in*R*_h_ of smaller aggregates. This demonstrates the protective effect of HPCD and Bet. To further investigate these effects, we needed other, more direct methods for the study of protein stability and oligomeric state.

### 2.2. Effect of the Chemical Chaperones on the Oligomeric State of GDH Denatured by the Elevated Temperature or a Freeze–Thaw Cycle

It is known that aggregate sizes cover a range from small oligomers to visible “snow” and precipitates, and generally only the smaller species are reversible [39]. It is these small reversible aggregates and the chemical chaperones effect on them that we have studied by the AUC method, while large aggregates we have studied by DLS. Figure 3A shows the sedimentation behavior of the protein pre-heated with or without the chemical chaperones for 10 min at 50 °C, then quickly cooled and placed into the rotor cells for the AUC run. Up to 68% of the protein form insoluble aggregates in the absence of cosolutes and precipitate during the rotor acceleration. The analysis of the *c*(*M*) distribution (Figure 3A, black curve) shows that the remaining protein is a mixture of hexamers (a shoulder at 389 ± 31 kDa) and higher molecular weight associates (a peak at 487 ± 40 kDa and a shoulder at 638 ± 30 kDa). There is also a minor peak (121 ± 18 kDa), that corresponds to the dimer of GDH (Figure 3A). 300 mM Sorb, 500 mM Tre or 1 M Bet completely protected GDH from the precipitation (the fraction of the protein aggregates precipitated during the rotor acceleration, γ_agg_, is 0%), while 350 mM Tre or 100 mM HPCD protected the protein only partially (γ_agg_ = 18% and 30% respectively). In the presence of the chemical chaperones the distributions are quite broad. However, 300 mM Sorb stabilizes the main hexameric form of GDH (narrow peak at 384 ± 9 kDa) and its small associates with a mass of 466 ± 8 kDa (Figure 3A, orange curve). HPCD also stabilizes the hexameric form of GDH (peak at 344 ± 17 kDa) and its small associates (peak at 430 ± 28 and minor peak at 610 ± 29 kDa) (Figure 3A, blue curve). 500 mM Tre stabilizes larger associates/aggregates as compared to Sorb (Figure 3A, green curve) and a small amount of dimeric form of GDH with the mass of 132 ± 18 kDa (Figure 3A, green curve). Thus, Tre can stabilize the intermediate (dissociative) form of GDH. 1 M Bet stabilizes the largest forms of the protein with masses ranging from 500 to 1170 kDa (Figure 3A, red curve).

Figure 3B shows the *c*(*M*) mass distribution for GDH frozen in the presence of the chemical chaperones (350 mM of Tre, 100 mM of HPCD, 100 mM of Sorb). There is no control distribution for GDH_fr_ without chaperones since almost all the protein precipitates after thawing (see a gray arrow in Figure 3C). All the chemical chaperones protected GDH_fr_ from the precipitation (γ_agg_ = 0, Figure 3C) at the given concentrations. The *c*(*M*) distribution for GDH_fr_ in the presence of 350 mM Tre revealed two maxima at 465 ± 31 kDa and 596 ± 26 kDa. In the presence of 100 mM HPCD the *c*(*M*) distribution revealed two peaks at 509 ± 21 kDa and 643 ± 27 kDa. GDH_fr_ in the presence of 100 mM Sorb had a more polydisperse distribution in the mass range from 380 to 750 kDa and with small peaks at 828 kDa and 243 kDa. The polydispersity of the distributions indicates the presence of various oligomeric forms and conformational states of GDH_fr_. Thus, all the chemical chaperones preferably stabilized relatively small soluble GDH_fr_ associates/aggregates.

It should be emphasized that the majority (95%) of GDH precipitated after a freeze–thaw cycle. As can be seen from Figure 3C, the presence of small concentrations of HPCD (0.001 to 1 mM) or Tre (0.05 to 10 mM) during GDH freezing does not protect the protein from aggregation. A significant stabilizing effect was obtained only at higher concentrations of the chemical chaperones. If the freezing of GDH occurred in the presence of 50 mM HPCD, then almost 3-foldless protein precipitated compared to the control sample, and no GDH_fr_ precipitation occurred at HPCD concentrations ranging from 75 mM to 175 mM (γ_agg_ = 0). In the case of Tre, a protective effect of the chemical chaperone became evident at 25 mM Tre and then essentially increased at 75–150 mM Tre. Upon freezing with 200–350 mM Tre, GDH did not precipitate (γ_agg_ = 0). Figure 3C also shows that the concentrations of Sorb used (50–100 mM) completely protected GDH from precipitation after a freeze–thaw cycle. More detailed data on the protective effect of the chemical chaperones are presented in Appendix A (Appendix A).

### 2.3. Effect of the Chemical Chaperones on the Thermostability of the Intact Proteins and after a Freeze–Thaw Cycle

To study the effect of the chemical chaperones on the protein thermal stability, DSC was used. The temperature dependences of the excess heat capacity for GDH (0.4 mg/mL) in the absence or presence of the chemical chaperones are shown in Figure 4A. Table 2 presents the main parameters characterizing the thermal unfolding of the protein: the position of the DSC curve maximum (*T*_max_) and the thermal transition enthalpy (Δ*H*_cal_), defined as the area under the curve. For GDH (Figure 4A, black curve), the *T*_max_ value corresponds to 51.8 °C and shifts towards higher temperatures by 0.9–3.9 °C upon the addition of 500 mM Bet, 350 mM Tre, 75 mM HPCD, or 1 M Sorb (Table 2, column with Δ*T*_max_ values). It can be concluded that the thermal stability of GDH increases under the action of the chemical chaperones, which is especially significant in the presence of 1 M Sorb. According to Table 2, the Δ*H*_cal_ value for GDH remains almost unchanged in the presence of Bet, Tre, or Sorb, but decreases significantly in the presence of HPCD. It should be noted that similar data on the decrease in the GDH enthalpy under the action of HPCD were obtained by us earlier on another GDH preparations and on another calorimeter (DASM-4M, Pushchino, Russia). This indicates that HPCD stimulates a conformational change in the GDH molecule, and its transition from the native to the denatured state requires less energy.

Figure 4B shows that the freeze–thaw cycle results in the loss of the tertiary structure of the GDH molecule and the absence of a cooperative thermal transition in the protein melting (black curve). Chemical chaperones can prevent this process and stabilize the protein structure during freezing. It was shown that the thermal stability of GDH frozen in the presence of 50 mM Tre or Sorb is comparable to that of the native protein, but similar concentrations of HPCD and Bet lead to a shift of the maximum of the melting curve towards higher temperatures (Figure 4B; Table 2, *T*_max_ values). The DSC profiles for GDH_fr_ in the presence of Tre/Sorb or Bet/HPCD are almost identical. It is interesting to note that Δ*H*_cal_ values for GDH_fr_ + chaperone are very similar for different agents (Table 2).

To test whether the freeze–thaw cycle affects proteins in a similar way, and to evaluate the effect of the chemical chaperones on a protein under these conditions, DSC studies were carried out for another enzyme, glycogen phosphorylase *b* (Ph*b*; 0.03M Hepes buffer, pH 6.8, containing 0.1 M NaCl). It has been previously shown that Bet and Tre increase the thermal stability of Ph*b* [36], while HPCD has a destabilizing effect on the protein [29]. Figure 4C shows the DSC profiles for the intact Ph*b* in the absence and presence of 500 mM Bet, 350 mM Tre, or 75 mM HPCD, which are consistent with those earlier obtained.

The DSC curves obtained for Ph*b* frozen at −20 °C (Ph*b*_fr_) in the absence or presence of the chemical chaperones and thawed at room temperature are presented in Figure 4D. It has been shown that Ph*b*_fr_ does not lose its tertiary structure during a freeze–thaw cycle (black curve). However, a slight decrease in the protein thermal stability is observed, as evidenced by a shift in the DSC profile from 56.5 °C to 55.8 °C (*T*_max_ for Ph*b* and Ph*b*_fr_, respectively). According to Table 2, the *T*_max_ positions for Ph*b*_fr_ in the presence of 500 mM Bet or 75 mM HPCD are similar to those obtained for Ph*b*, while 350 mM Tre stabilizes Ph*b*_fr_ less effectively than the intact protein. The Δ*H*_cal_ value for Ph*b*_fr_ in the absence or presence of the chemical chaperones changes little compared to similar values for the intact protein. This indicates that there are no significant structural rearrangements in the protein molecule as a result of freezing–thawing.

### 2.4. Effect of the Chemical Chaperones on the Stability of Phb Denatured by the Elevated Temperature or a Freeze–Thaw Cycle

The effect of the chemical chaperones on the oligomeric state of Ph*b* during thermal denaturation was studied by the AUC method. About 34% of the protein form insoluble aggregates in the absence of cosolutes and precipitate during the rotor acceleration (Table 3). The remaining protein sediments as a mixture of non-native dimers and tetramers (Figure 5). The osmolytes, Bet and Tre, stabilize the protein, reducing significantly the values of γ_agg_ up to 9% in the presence of 500 mM Bet or 300 mM Tre (see Table 3), while the presence of 75 mM HPCD slightly destabilizes the protein (γ_agg_ = 40%).

The *c*(*M*) distribution for Ph*b* after 40 min heating at 48 °C followed by rapid cooling reveals the main peak at 255 ± 9 kDa and two minor peaks at 363 ± 8 kDa and 176 ± 15 kDa, which indicate the presence of a mixture of probably dimers and tetramers and small aggregates (Figure 5). The addition of 300 mM Tre stabilizes the dimeric form of Ph*b* (197 kDa) (not shown since this coincides with our previously obtained data [36]). Tre can also stabilize denatured Ph*b* monomers and small oligomeric aggregates and thereby inhibit the formation of large aggregates [36]. The *c*(*M*) distribution for the pre-heated mixture of Ph*b* with 75mM HPCD shows one minor peak (184 ± 9 kDa) corresponding to the dimeric form of the protein, a major broad peak in the mass range of 220–382 kDa, and a small shoulder at 445 ± 30 kDa, which may correspond to the HPCD complex with a partially unfolded dimer and tetramer of Ph*b* and small aggregates. The *c*(*M*) distribution of Ph*b* heated in the presence of 300 mM Bet reveals the main peak at 489 kDa and two minor peaks at 383 and 592 kDa, indicating the presence of tetramers and relatively small aggregates of Ph*b* in the solutions.

When studying the effect of freezing at –20 °C and subsequent thawing on Ph*b*, it was shown that the protein forms aggregates. These aggregates are quite stable at 37 °C, judging by the DLS method. It can be seen from Figure 6A that the *I* value of the Ph*b* solution after a freeze–thaw cycle does not change much. The mass particle size distribution after 2 h incubation at 37 °C is shown in Figure 6B (dotted curve). The *R*_h_ values for most of the particles present in the solution are in the range of 9–100 nm with the distribution maximum (*R*_h,max_) at 14.6 ± 1.8 nm. A number of aggregates larger than 100 nm are also registered. It was shown that 38% of the protein can be precipitated during centrifugation, and the chemical chaperones (Bet, Tre, HPCD) diminish the amount of precipitated Ph*b*_fr_ (Table 3). The mass particle size distributions shift towards smaller radii and the *R*_h,max_ values equal 11.3 ± 1.6 nm and 5.8 ± 0.9 nm in the presence of Tre and Bet, respectively (Figure 6B). Despite a 3.5-fold decrease in the amount of Ph*b*_fr_ precipitated by centrifugation (cf. γ_agg_ values in Table 3), HPCD causes an increase in the size of aggregates formed in the Ph*b*_fr_ solution. It should be noted that Ph*b* destabilization and aggregation enhancement in the presence of HPCD was observed earlier upon thermal aggregation of Ph*b* [29]. Here, HPCD-induced aggregation/association of the intact Ph*b* at the physiological temperature has been also observed (blue dashed curve in Figure 6A). This process accelerates for the frozen protein (blue solid curve in Figure 6A). As can be seen from Figure 6C, HPCD leads to the formation of large particles up to 1 μm after 2 h incubation at 37 °C. Nevertheless, a sufficient fraction of particles of a lesser size with approximate <*R*_h_> = 4.5 ± 0.3 nm is also registered in the solution.

### 2.5. Effect of HPCD on the Stability of GDH and Phb Characterized by CD Spectroscopy

The comparison of CD spectra allows one to detect changes in the secondary structure of proteins and, therefore, their stability in the absence and presence of additives. The dual effect of HPCD on the aggregation and stability of the target proteins prompted us to investigate the effect of this compound on the secondary structure of GDH and Ph*b*. The CD spectra of GDH and Ph*b* in the absence and in the presence of HPCD are shown in Figure 7.

The calculation of the secondary structure of GDH from the CD spectra shows that despite its effect on the protein aggregation and DSC profiles (Figure 1A and Figure 4A), HPCD does not significantly change the content of structural elements in the GDH molecule. The addition of 1 mM HPCD causes a change of only 1% of α-helices, and 75 mM HPCD increases the content of α-helices by 4% (Table 4). The presence of 50 mM HPCD in the freeze–thawed sample of GDH preserves the secondary structure of the protein and also increases the α-helices content by 3%. The freeze–thawing of GDH without HPCD leads to the loss of the protein due to its precipitation (the black line in Figure 7A). In contrast with GDH, 75 mM HPCD induces significant changes in the secondary structure of Ph*b* (Figure 7B). The number of α-helices decreases by 10% (from 51 to 41 %), while the number of β-strands increases from 11 to 16 % (Table 4).

The CD data for Ph*b* indicate that the stability of this protein decreases already at room temperature in the presence of HPCD. At the same time, comparable concentrations of HPCD do not significantly disturb the secondary structure of GDH. The results also confirm the cryoprotective effect of HPCD on GDH.

## 3. Discussion

There are several criteria of protein stability estimation, for example, by changes in the activity of a protein or in its structure. In vivo protein stability is related to the proteome stability, and cell resistance to various stresses. Living cells have the proteostasis network, including chaperones, the mechanisms of protein disaggregation and refolding, and the accumulation of osmolytes under stress. Considering the importance of the absence of insoluble aggregates for the cell, this may be another criterion for protein stability.

In the present work we have focused on the effect of the osmolytes and HPCD on the structural stability at different levels of spatial organization and the formation of insoluble aggregates of two allosteric proteins upon thermal or freeze–thaw stress. Thermal aggregation of GDH and Ph*b* was chosen for the comparison with the freeze–thaw-induced aggregation of these proteins.

After freezing and thawing, almost all GDH denatures, aggregates, and precipitates (DSC, CD and AUC data). In the case of heating for 10 min at 50 °C intact GDH precipitates by 68% during centrifugation, and the remaining fraction of GDH exists as a mixture of hexamers and small associates/aggregates (Figure 3A). All the tested cosolutes (Tre, Sorb, Bet, HPCD) increase the thermal stability of GDH (Figure 4, Table 2), protect the protein from aggregation (Figure 1) and subsequent precipitation (Figure 3C and Appendix A). However, there are some differences in their action.

Previously, Kurganov suggested using the parameter [L]_0.5_ from the Hill equation as an estimation of the anti-aggregation activity of the chemical chaperones [40]. Judging from the [L]_0.5_ values (Table 1), HPCD and Tre appear to be the most effective during thermal aggregation of GDH, while HPCD and Bet—during the freeze–thaw cycle. The latter is supported by the DSC data, since the thermostability of GDH_fr_ in the presence of Bet or HPCD is higher than in the presence of other cosolutes (Table 2).

According to the DLS data on the kinetics of protein aggregation, sub-millimolar concentrations of HPCD (<1 mM) inhibit GDH aggregation after a freeze–thaw cycle (Figure 1C,D), but aggregates with *R*_h_ greater than 1 µm and aggregates of a much smaller size are registered (Figure 2). The formation of two populations of radii after freeze–thawing was also observed for human growth hormone (hGH) [41]. Eckhardt and co-authors suggested [41] that large insoluble aggregates formed during hGH freeze–thawing were not simply an extension in size or mass of soluble small-sized aggregates, and were produced by a different mechanism. The authors assumed that a possible explanation for the formation of large aggregates might be surface denaturation occurring at a relatively large ice-liquid interface between the protein and the tiny ice crystals that had been formed after freezing. These crystals could serve as sites for protein denaturation and the formation of large particles.

The AUC data show that the formation of rather large insoluble aggregates of GDH_fr_ ceases at much higher concentrations of HPCD (Appendix A). Therefore, direct method demonstrates that HPCD concentrations <1 mM provide a low level of protein protection, and even in the presence of 50 mM HPCD, there is still a significant fraction of GDH_fr_ forming large aggregates that precipitate during the rotor acceleration (Figure 3C, Appendix A). When comparing data from different studies, one should take into account the exact purpose of the studies, the methods used, and the criteria chosen to assess the protein stability. This means that if the preliminary comparison of these effects can be made by simpler techniques such as light scattering measurement or apparent absorption registration, then a more thorough and complex study by other methods may be required in each case. HPCD completely protects GDH_fr_ from precipitation at the concentrations of 75 mM and above, while GDH_fr_ is in the form of small associates/aggregates rather than native hexamers. In contrast, during the short-term thermal stress (10 min at 50 °C) the presence of even 100 mM HPCD does not completely protect GDH from precipitation. Earlier Sabbaghian and colleagues [42] showed that cyclodextrins could not preserve GDH native structure upon heating, and suggested that cyclodextrins might stabilize an intermediate/unfolded state of the protein, thereby protecting it from further aggregation. Our data also indicate this effect for HPCD, since both small associates/aggregates and forms with a mass near the GDH hexamer are detected in this case (Figure 3A).

It has been previously shown that low concentrations of Tre stabilize some proteins, whereas high concentrations of Tre have a destabilizing effect (see review [21]). On the contrary, our data obtained by DLS and AUC methods show that low concentrations of Tre (0.5–10 mM) have an insignificant protective effect on GDH after a freeze–thaw cycle, the protein easily aggregates (Figure 1D) and precipitates. The protective effect increases with increasing Tre concentration, and the complete prevention of aggregation occurs at 200 mM Tre (Figure 3A), while the protection under heat stress requires 500 mM Tre. We have also shown that Tre can stabilize the intermediate/dissociated form of GDH during heat stress (Figure 3A). According to the AUC data, upon both heat– and freeze–thaw-induced denaturation in the presence of Tre, GDH forms a polydisperse set of associates/aggregates (Figure 3A,B) with masses ranging from 370 to 700 kDa. It is known that trehalose protects proteins against freeze–drying damage in cells [43]. It was reported that trehalose generally protected proteins against three types of stress: freeze–thaw, freeze-drying and vacuum-drying. Tre protected lactate dehydrogenase (LDH) activity against freeze–thaw stress (see review [44]), at least partially by the enhanced hydrogen bonding between trehalose and proteins [44,45].

According to the AUC data, Sorb is more effective in protection of the quaternary structure of GDH_fr_, than other cosolutes. 50 mM Sorb completely protects freeze–thawed GDH_fr_ from precipitation, although 300 mM Sorb is needed to protect the protein under heat stress. According to the AUC data, small associates/aggregates with a mass of 380–850 kDa are formed during freezing, and the intermediate dimeric form is stabilized (at 100 mM of Sorb). During thermal aggregation of GDH, we observe two narrow peaks with mass of 384 kDa (hexameric form) and 466 kDa (aggregates of small intermediate forms).

Bet demonstrates the least protective effect on the thermal stability of intact GDH (DSC data, ∆*T*_max_ = 0.9 °C, Table 2) and the lowest anti-aggregation activity (DLS data, [L]_0.5_ = 384 ± 11 mM, Table 1) under heat stress. In the presence of 1 M Bet thermally denatured GDH has the widest polydisperse distribution *c*(*M*), shifted towards higher molecular weights in the range of 500–1200 kDa (AUC data, Figure 3A). It should be noted that at such concentrations of Bet, a significant contribution can be made by the molecular crowding/excluded volume effect. However, under freeze–thaw stress, Bet effectively influences the stability of the model protein at the much lower concentration. 50 mM Bet increases the thermostability of GDH_fr_ by shifting the maximum of thermal transition by 2.2 °C (Figure 4B, Table 2) towards the higher temperatures and completely prevents aggregation (Figure 1, Table 1). At the same time, 1M Bet partially protects phosphofructokinase but does not protect LDH or lipase against freeze drying [44].

The above data suggest that lower concentrations of cosolutes (HPCD, Tre, Sorb, Bet) protect GDH from the formation of large aggregates during its freeze–thawing than during heating. It should be noted that [L]_0.5_ values for each agent are also significantly lower for aggregation of frozen–thawed GDH than for thermal aggregation of intact GDH. The difference is more than an order of magnitude (Table 1). It can be assumed that the rate of aggregation of the denatured protein is important here, which also depends on the viscosity of the solutions. At lower temperatures, the viscosity of the solution increases, and the lower concentration of cosolutes is sufficient to slow down the aggregation process.

According to the AUC data, GDH_fr_ stabilized by Tre, Bet, Sorb, or HPCD is in the form of small associates (GDH–GDH suboligomers or GDH–chaperone complexes) with a mass of about 380–700 kDa. The stabilization of oligomeric GDH associates by chemical chaperones (proline, arginine and its derivatives) was demonstrated earlier during its thermal aggregation [32] and we affirmed this for the other four chemical chaperones used in the present work (Figure 3A). The results obtained in the present work show that the sizes of both small- and large-sized aggregates diminish in the presence of chemical chaperones. Thus, the stabilization of soluble small-sized GDH aggregates appears to be a common feature of the protective action of the chemical chaperones independent of whether the protein has been denatured by heat or freeze–thaw. Thereby, these chemical chaperones prevent large-scale aggregation, even if they are not able to preserve the protein in its native form.

The similar stabilization effect of Bet and Tre on thermal aggregation of Ph*b* was shown by us earlier [35,36,37,38]. In the current work, we have also investigated the effect of these cosolutes during freeze–thawing of Ph*b*. It was shown that the freeze–thaw cycle slightly destabilizes the protein (Figure 4D), Ph*b* molecules form a polydisperse set of aggregates (Figure 6B) rather stable at 37 °C. When Ph*b* is frozen with Bet or Tre, the particle size distribution shifts towards smaller sizes (Figure 6B), and the amount of insoluble aggregates decreases (Table 3). Thus, our data indicate that smaller Ph*b*_fr_ aggregates are stabilized in the presence of Bet and Tre resulting in the aggregation slowdown, as in the case of GDH. This may be due to a change in the mechanism of protein aggregation in the presence of these cosolutes.

The destabilizing effect of HPCD on Ph*b* was observed both for the intact protein and for the protein after the freeze–thaw cycle (Figure 4C,D and Figure 6A). This is not the only known case when cyclodextrins destabilize proteins. HPCD promotes the destabilization and aggregation of glyceraldehyde-3-phosphate dehydrogenase [30] or lysozyme [28]. In some cases, high and low concentrations of HPCD have different effects on the stability of the same protein. HPCD increases the melting temperature (*T*_max_) of recombinant porcine growth hormones [46] in the presence of low (18 mM) concentration. However, in the presence of its high concentration (180 mM), the onset temperatures of the protein unfolding and aggregation decrease, becoming 2.5–3.5 °C lower.

Since HPCD has an ambiguous effect on the stability of the model proteins in our work, its effect on the protein secondary structure has been studied. Earlier it was shown that 10-min heating at 50 °C resulted in the destruction of GDH secondary structure and cyclodextrins could not preserve it [42]. According to the obtained data, after freeze–thawing in the absence of HPCD GDH completely loses its secondary structure either (Figure 7). Nonetheless, in the presence of 50 mM HPCD in the solution the frozen–thawed GDH retains its secondary structure. Moreover, it has been shown that HPCD can even stimulate GDH helication: the number of α-helices in the intact GDH increases by 6% in the presence of 150 mM HPCD at room temperature (Table 4). On the contrary, in the case of the intact Ph*b*, HPCD reduces the degree of protein helication by 10% even at room temperature, while the number of β-structures increases (Table 4). A similar effect was observed for the antidiabetic α-incretin agonist (MEDI0382), which contained a high amount of α-helices (51%) and few β-structures (11%). In the presence of HPCD it exhibited a helicity loss of up to 18% compensated by an increase in the β-sheet and random coils [47]. Apparently, these changes in the Ph*b* secondary structure under the action of HPCD lead to the lower thermal stability of the protein (Figure 4C), observed aggregation/association of the intact protein (Figure 6A), and a higher fraction of aggregated protein after heating at 48 °C (Figure 5, Table 3).

## 4. Materials and Methods

### 4.1. Materials

Bovine liver GDH (suspension in ammonium sulfate), mono- and dibasic sodium phosphates, sodium chloride, Hepes were purchased from Sigma-Aldrich (St. Louis, MO, USA). Betaine was purchased from ICN Biomedicals (Costa Mesa, CA, USA). NaCl was purchased from Reakhim (Moscow, Russia). All solutions for the experiments were prepared using deionized water obtained with the Arium Mini system (Sartorius, Göttingen, Germany) and Simplicity UV system (Millipore, Burlington, MA, USA).

For experiments with GDH, all samples and stock solutions of the chemical chaperones were prepared in 0.1 M Na-phosphate buffer, pH 7.6, containing 0.01 M NaCl. For experiments with Ph*b* (except for AUC and CD), samples and stock solutions of the chemical chaperones were prepared in 0.03 M Hepes buffer, pH 6.8, containing 0.1 M NaCl.

### 4.2. Protein Preparations

GDH was purified from the suspension using PD-10 desalting column (GE Healthcare, Little Chalfont, Buckinghamshire, UK) according to the instructions. The procedure for Ph*b* isolation from rabbit skeletal muscle was carried out as described in [48]. Protein concentrations were determined using Spekol 1300 spectrophotometer (Analytik Jena, Jena, Germany) or NanoPhotometer NP80 (Implen, München, Germany), using the extinction coefficients of 1 mg/mL protein (*A*_0.1%_^280^) 0.97 for GDH [49] and 1.32 for Ph*b* [50].

### 4.3. Dynamic Light Scattering (DLS)

The aggregation kinetics was registered by an increase in the light scattering intensity using the Photocor Complex correlation spectrophotometer (PhotoCor Instruments, Inc., College Park, MD, USA) with a He-Ne laser (Model 31-2082, 632.8 nm, 10 mW, Coherent Inc., Santa Clara, CA, USA) as a light source. The scattered light was collected at 90° angle; the accumulation time of autocorrelation functions was 30 s for each experimental point. All experiments were repeated in triplicate.

In the case of heat-induced aggregation, glass vials with stoppers were pre-heated at the desired temperature (50 °C for GDH or 37 °C for Ph*b*) with the buffer and with/without a chemical chaperone. The aggregation experiments were initiated by adding the protein into the pre-heated solution to a final concentration of 0.2 mg/mL for GDH or 0.3 mg/mL for Ph*b*.

In the case of freeze–thaw-induced aggregation, the samples with 0.2 mg/mL GDH or 0.3 mg/mL Ph*b* were frozen with or without chemical chaperones in plastic centrifuge microtubes at −20 °C for at least 12 h. The samples were defrosted just before the experiment in a thermostat at 20 °C to equalize the conditions of thawing. After complete thawing the samples were placed in glass vials pre-heated at the desired temperature (25 °C for GDH_fr_ or 37 °C for Ph*b*_fr_), and then the aggregation kinetic curves were registered.

The initial parts of the kinetic curves were analyzed using the equation [40]:(1)$$I=I_{0}+v_{0}\left(t-t_{0}\right)-B{\left(t-t_{0}\right)}^{2}$$
where *I*_0_ is the initial value of the light scattering intensity, *v*_0_ is the parameter characterizing the initial aggregation rate at the stage of aggregates growth, *t*_0_ is the duration of the lag period, determined by the segment on the abscissa, cut off by the theoretical curve calculated from this equation, and B is a constant.

The anti-aggregation activity of the chaperone was characterized by the half-saturation concentration [L]_0.5_ [40]:(2)$$\frac{v_{0,add}}{v_{0}}=\frac{1}{{1+\left(\frac{\left[L\right]}{{\left[L\right]}_{0.5}}\right)}^{h}}$$
where *v*_0,add_ is the parameter characterizing the initial aggregation rate in the presence of the additive studied, [L] is the concentration of the additive, *h* is a coefficient.

The *R*_h_ values and the mass/number particle size distributions were calculated using DynaLS software v.2 (Alango, Tirat Carmel, Israel) and the specialized algorithm freely available at https://dls.rogach.org (accessed on 11 April 2023) as described in [51].

### 4.4. Analytical Ultracentrifugation (AUC)

Sedimentation velocity (SV) experiments were carried out using a Model E analytical ultracentrifuge (Beckman Instruments, Palo Alto, CA, USA) with absorbance optics, a photoelectric scanner, a monochromator and a computer online. A four-hole rotor An-F Ti and 12-mm double sector cells were used. The rotor was pre-heated in a thermostat at 25 °C overnight before the run.

In the case of GDH SV runs were performed at 25 °C in 0.1 M Na-phosphate buffer, pH 7.6, containing 0.01 M NaCl, with or without chemical chaperones. GDH samples were incubated for 10 min at 50 °C or frozen overnight and defrosted at 20 °C before the run. In the case of Ph*b* the SV experiments were performed in a 0.03 M HEPES buffer, pH 6.8, containing 0.15 M NaCl, with or without chemical chaperones. All Ph*b* samples were incubated for 40 min at 48 °C, and then cooled for 2 min in ice water, introduced into the cells, and experiments were carried out at 25°C. The rotor speed for all SV experiments was 48,000 rpm.

Sedimentation profiles were recorded by measuring the absorbance at 280 nm. All cells were scanned simultaneously. The time interval between scans was 2.5 min. The differential sedimentation coefficient distributions [*c*(*s*) versus *s* or *c*(*M*) versus *M*] were determined using SEDFIT program (v. 14.0 and higher) [52]. Sedimentation coefficients were corrected to the standard conditions (a solvent with the density and viscosity of water at 20 °C) using SEDFIT [52]. The values of the density and the dynamic viscosity of the species used in the AUC measurements at 25 °C are presented in Appendix A of the Appendix A.

### 4.5. Densitometry and Viscosimetry

The dynamic viscosities (η) of the buffer and chemical chaperones solutions were measured using the AMVn microviscometer (Anton Paar, Graz, Austria) with 1.6/1.5 mm capillary system. The densities (ρ) of these solutions were measured using the DMA 4500 densitometer (Anton Paar, Graz, Austria). The refraction indices (*n*) of all solutions were measured using the ABBEMAT 500 refractometer (Anton Paar, Graz, Austria). All measurements were made at 25 and 37 °C. The obtained values of parameters are given in the Appendix A (Appendix A).

### 4.6. Differential Scanning Calorimetry (DSC)

The experiments were performed using a MicroCal VP-Capillary DSC differential scanning calorimeter (Malvern Instruments, Northampton, MA, USA). The heating rate was 1 °C/min. The protein concentration was 0.4 mg/mL for GDH, 0.8 mg/mL for GDH_fr_ and 1 mg/mL for Ph*b* and Ph*b*_fr_. The experiments were conducted in 0.1 M Na-phosphate buffer, 0.01 M NaCl, pH 7.6 for GDH/GDH_fr_ and in 0.03 M Hepes buffer, 0.1 M NaCl, pH 6.8 for Ph*b*/Ph*b*_fr_. For experiments with GDH and Ph*b*, the intact protein at the selected concentration was used. In the experiments with chemical chaperones, the same concentration of a chaperone was added to both control and sample cells. For experiments with GDH_fr_ and Ph*b*_fr_, the selected concentrations of protein were frozen with or without cosolutes at −20 °C for at least 12 h and thawed in a thermostat at 20 °C just before the addition to the sample cell. A buffer solution containing the same concentrations of additives and subjected to a similar freeze–thaw cycle was used in the control cell. Additional measurements of the protein concentration in the sample after thawing were not carried out. The correction of the calorimetric traces, analysis of the temperature dependence of the excess heat capacity, the thermal stability estimation and calculation of the calorimetric enthalpy (Δ*H*_cal_) were performed as described in [53].

### 4.7. Circular Dichroism (CD) Spectroscopy

CD spectra of Ph*b* or GDH (0.2 mg/mL) were recorded in the wavelength interval 190–280 nm on the Chirascanspectropolarimeter (Applied Photophysics Ltd., Surrey, UK) equipped with a thermoelectric temperature control unit. The spectra were recorded in 0.5 mm cells in 0.1 M Na-phosphate buffer, pH 7.6 at 25 °C for GDH and in 0.015 M Hepes buffer, 0.1 M NaCl, pH 6.8 for Ph*b* at 20 °C. For experiments with GDH_fr_, the selected concentration of the protein was frozen in the presence of 50 mM HPCD at −20 °C for at least 12 h and thawed in a thermostat at 20 °C just before the addition to the sample cell. Additional measurements of the protein concentration in the sample after thawing were not carried out. The results were a mean of 5 scans. Blanks with a buffer or a solution of HPCD at the concentrations used in the experiments were recorded for all samples and subtracted accordingly. The scanning speed was 20 nm/min. The spectral band width was equal to 1 nm. Protein secondary structure determinations were obtained using the DICHROWEB server [54] (accessed on 17 March 2023) with the CDSSTR analysis program and assuming SP180 as the dataset reference [55].

### 4.8. Calculations

The Origin software (OriginLab Corporation, Northampton, MA, USA) version 8.0 and higher was used to analyze the obtained data. DSC data analysis was carried out using MatLab (The MathWorks, Inc., Natick, MA, USA), version R2015a.

## 5. Conclusions

The stability of proteins as well as the factors influencing it and suppressing protein aggregation are important for the therapeutic strategy of conformational disease prevention and treatment. When characterizing the protective effect of various agents, including chemical chaperones, one should take into account that anti-aggregating agents may have different effects under different conditions (physical stress factors, concentration of the studied agent, solution composition, molecular crowding). In this work, we have studied the effect of two types of stress, heat and freeze–thawing, on two oligomeric allosteric enzymes Ph*b* and GDH. It was shown that both stresses slightly differed in their effect on Ph*b* structure, and in the case of GDH, freeze–thaw stress had a much stronger effect.This work showed that freezing–thawing almost completely destroyed the secondary and tertiary structure of GDH, but chemical chaperones (Tre, Bet, Sorb, HPCD) could prevent this even at relatively low concentrations. Our results may contribute to the fundamental research on understanding the effect of chemical chaperones on protein stability and aggregation under different kinds of stresses.

## Figures and Tables

**Figure 1 ijms-24-10298-f001:**
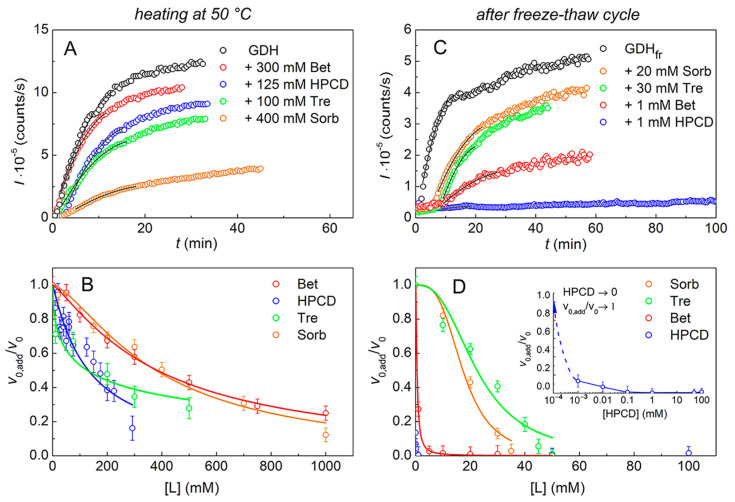
The kinetics of GDH aggregation (0.2 mg/mL, 0.1 M Na-phosphate buffer, pH 7.6, 10 mM NaCl) induced by thermal stress at 50 °C and after freeze–thawing at 25 °C. (**A**,**C**) The aggregation kinetic curves *I*(*t*) for GDH or GDH_fr_, respectively, in the absence or presence of HPCD, Tre, Sorb, or Bet. The additive concentrations are shown in the panels. Points are the experimental data, lines are the fitting curves calculated with Equation (1). (**B**,**D**) The dependences of *v*_0,add_/*v*_0_ on the additive concentration, [L], for aggregation of GDH or GDH_fr_, respectively. *v*_0_ is the parameter representing the initial aggregation rate in the absence of the additives; *v*_0,add_—in the presence of the additives. The inset shows *v*_0,add_/*v*_0_ values at the micromolar concentrations of HPCD in the log scale. At zero concentration of additives the initial value of *v*_0,add_/*v*_0_ = 1.

**Figure 2 ijms-24-10298-f002:**
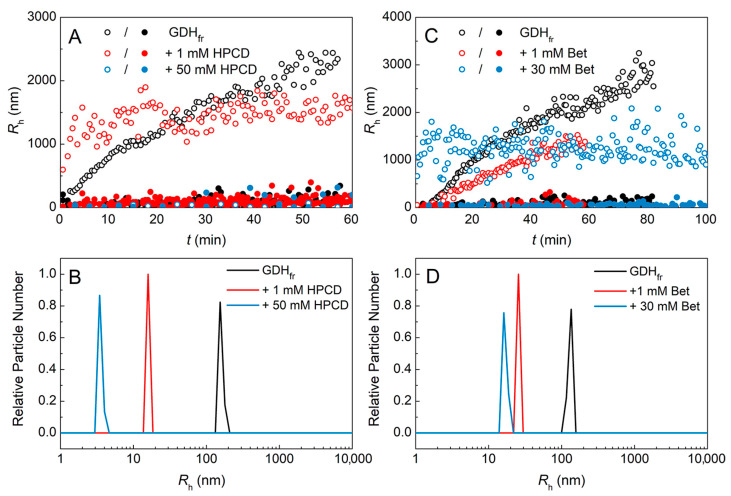
The average hydrodynamic radii (*R*_h_) of GDH_fr_ aggregates at 25 °C after freeze–thawing in the presence of HPCD or Bet. (**A**,**C**) The time dependences of *R*_h_ in the absence or presence of different concentrations of HPCD or Bet, respectively. The open circles indicate the populations (1) of aggregates with higher impact on the light scattering intensity (*I*), the closed circles—populations (2) of aggregates with little effect on *I*. (**B**,**D**) The distribution of particle number on *R*_h_ of GDH_fr_ in the absence and presence of HPCD or Bet, respectively, at 38 or 41 min after thawing.

**Figure 3 ijms-24-10298-f003:**
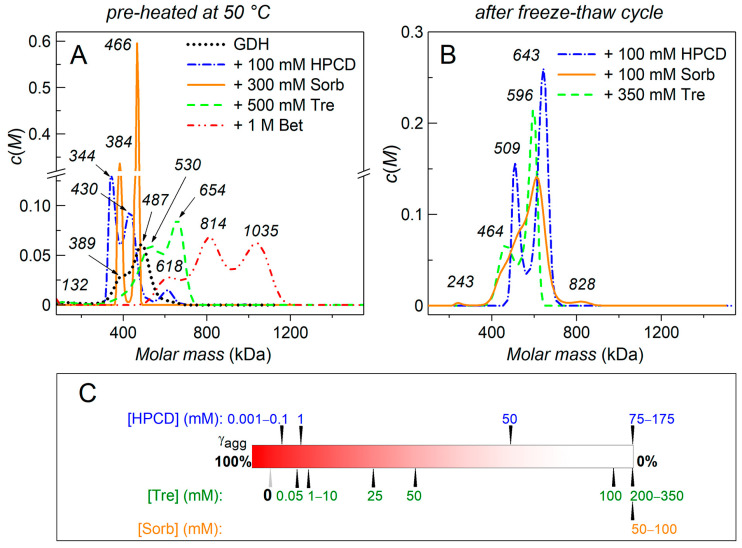
The effect of the chemical chaperones on the oligomeric state of GDH (0.2 mg/mL). (**A**) The differential *c*(*M*) distributions for GDH pre-heated for 10 min at 50 °C in the absence (black dotted curve) or presence of 100 mM HPCD (blue dot-dashed curve), 300 mM Sorb (orange solid curve), 500 mM Tre (green dashed curve), or 1 M Bet (red dash-dot-dotted curve). (**B**) The differential *c*(*M*) distributions for GDH after a freeze–thaw cycle in the presence of 100 mM HPCD (blue dot-dashed curve), 100 mM Sorb (orange solid curve), or 350 mM Tre (green dashed curve). The frozen samples were thawed at 20 °C before the run. All the distributions were obtained at 25 °C; the rotor speed was 48,000 rpm. (**C**) The evaluation of the precipitated fraction of protein aggregates (γ_agg_) of GDH frozen overnight at −20 °C in the absence or presence of various concentrations of HPCD, Tre, or Sorb, which precipitates during a rotor acceleration.

**Figure 4 ijms-24-10298-f004:**
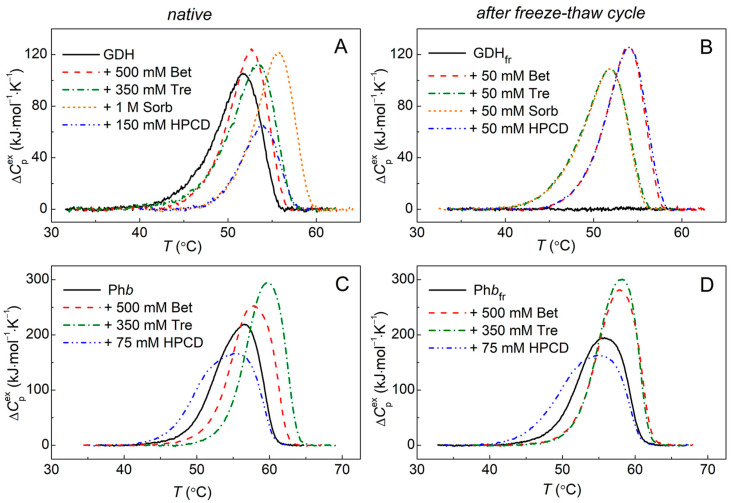
The effect of the chemical chaperones on the thermal stability of the intact and the frozen GDH and Ph*b*. The temperature dependences of the excess heat capacity ΔC_p_^ex^ for GDH (0.4 mg/mL, **A**), GDH_fr_ (0.8 mg/mL, **B**), Ph*b* (1 mg/mL, **C**) and Ph*b*_fr_ (1 mg/mL, **D**) in the absence of additives (black solid curves) or in the presence of Bet (red dashed curves), Tre (green dash-dotted curves), HPCD (blue dash-dot-dotted curves) or Sorb (orange dotted curves). Chaperone concentrations are shown on the panels. Ph*b*_fr_ and GDH_fr_ preparations were obtained by freezing the intact proteins at −20 °C in the absence or presence of the chemical chaperones. The samples were thawed at 20 °C before the experiment. DSC experiments were performed in 0.1 M Na-phosphate buffer, pH 7.6, 10 mM NaCl for GDH and GDH_fr_ and in 0.03 M Hepes buffer, 0.1 M NaCl, pH 6.8 for Ph*b* and Ph*b*_fr_.

**Figure 5 ijms-24-10298-f005:**
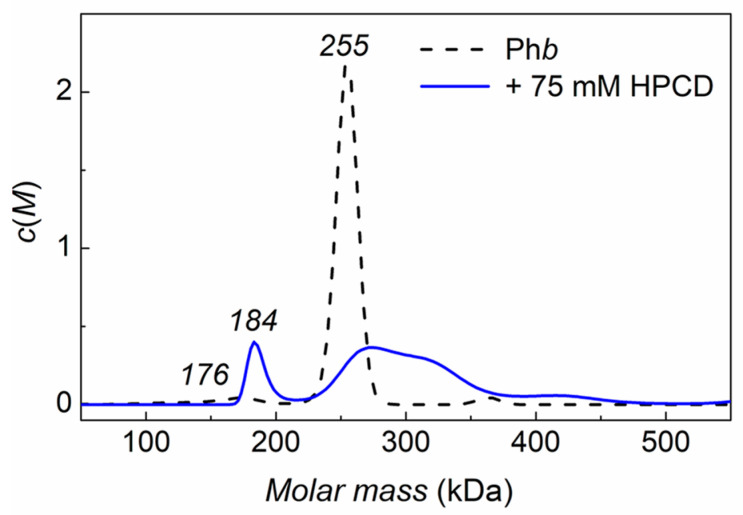
The effect of HPCD on the oligomeric state of thermally denatured Ph*b*.The *c*(*M*) distributions for 0.35 mg/mL Ph*b* (0.03 M Hepes buffer, pH 6.8, containing 0.15 M NaCl) pre-heated 40 min at 48 °C in the absence (black dashed curve) or presence of 75 mM HPCD (blue solid curve). All *c*(*M*) distributions were obtained at 25 °C. The rotor speed was 48,000 rpm.

**Figure 6 ijms-24-10298-f006:**
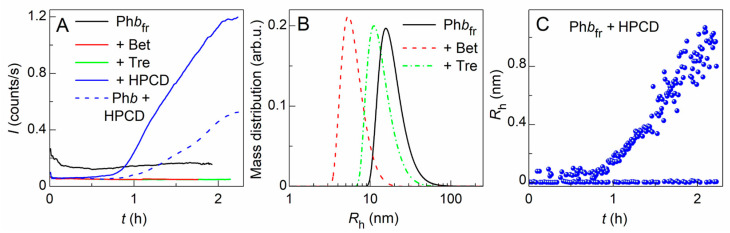
The effect of the chemical chaperones on the Ph*b* stability after the freeze–thaw cycle registered by DLS at 37 °C. (**A**) The dependences of the light scattering intensity (*I*) on time (*t*) for the thawed Ph*b* (0.3 mg/mL) after being frozen at –20 °C in the absence (Ph*b*_fr_, black curve) or presence of 350 mM Tre (green curve), 500 mM Bet (red curve), or 75 mM HPCD (blue solid curve), and the intact Ph*b* in the presence of 75 mM HPCD (blue dashed curve). (**B**) The Ph*b*_fr_ mass particle size distribution after 2 h incubation at 37 °C in the absence (black dotted curve) or presence of 350 mM Tre (green solid curve) or 500 mM Bet (red dashed curve). (**C**) The *R*_h_(*t*) dependence for Ph*b*_fr_ in the presence of 75 mM HPCD.

**Figure 7 ijms-24-10298-f007:**
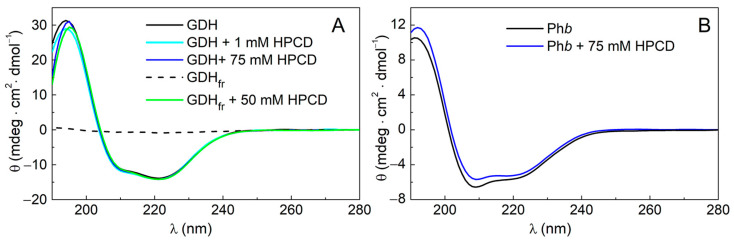
The effect of HPCD on the secondary structure of GDH and Ph*b*. (**A**) The CD spectra for GDH (0.2 mg/mL; 0.1 M Na-phosphate buffer, pH 7.6) in the absence of HPCD, in the presence of 1 mM HPCD, 75 mM HPCD, and for GDH frozen in the absence or in the presence of 50 mM HPCD. (**B**) The CD spectra for Ph*b*(0.2 mg/mL; 0.015 M Hepes buffer, 0.1 M NaCl, pH 6.8) in the absence or presence of 75 mM HPCD.

**Table 1 ijms-24-10298-t001:** The values of [L]_0.5_ and *h* for chemical chaperones suppressing thermal and freeze–thaw-induced aggregation of GDH.

Test-System	HPCD	Tre	Sorb	Bet
		**[L]_0.5_, mM**		
GDH at 50 °C	131 ± 13	126 ± 35	378 ± 24	384 ± 11
GDH_fr_ at 25 °C	<1	23 ± 2	17.3 ± 0.9	0.53 ± 0.02
		** *h* **		
GDH at 50 °C	1.1 ± 0.1	0.5 ± 0.1	1.5 ± 0.2	1.2 ± 0.1
GDH_fr_ at 25 °C	-	2.7 ± 0.6	3.3 ± 0.5	1.5 ± 0.1

**Table 2 ijms-24-10298-t002:** The main calorimetric parameters obtained for the thermal transitions of the intact and the frozen GDH and Ph*b* in the presence of the chemical chaperones.

Sample		*T*_max_, °C	Δ*T*_max_, °C *	Δ*H*_cal_, kJ⋅mol^−1^
GDH (0.4 mg/mL)	no additives	51.8 ± 0.1	0	669 ± 33
	+500 mM Bet	52.7 ± 0.1	+0.9	700 ± 35
	+350 mM Tre	53.4 ± 0.1	+1.6	715 ± 40
	+1 M Sorb	55.7 ± 0.1	+3.9	682 ± 35
	+150 mM HPCD	54.0 ± 0.1	+2.2	351 ± 28
GDH_fr_ (0.8 mg/mL)	no additives	–	–	–
	+50 mM Bet	54.0 ± 0.1	+2.2	700 ± 36
	+50 mM Tre	51.9 ± 0.1	+0.1	704 ± 41
	+50 mM Sorb	51.9 ± 0.1	+0.1	685 ± 35
	+50 mM HPCD	54.1 ± 0.1	+2.3	718 ± 40
Ph*b* (1 mg/mL)	no additives	56.5 ± 0.1	0	1665 ± 82
	+500 mM Bet	57.8 ± 0.1	+1.3	1818 ± 90
	+350 mM Tre	59.7 ± 0.1	+3.2	1915 ± 94
	+75 mM HPCD	55.0 ± 0.1	−1.5	1626 ± 75
Ph*b*_fr_ (1 mg/mL)	no additives	55.8 ± 0.1	−0.7	1585 ± 78
	+500 mM Bet	57.9 ± 0.1	+1.4	1964 ± 95
	+350 mM Tre	58.2 ± 0.1	+1.7	1996 ± 91
	+75 mM HPCD	54.9 ± 0.1	−1.6	1730 ± 85

* The Δ*T*_max_ column indicates the shift in the position of *T*_max_ in the presence of cosolutes and after the freeze–thaw cycle relative to the *T*_max_ of the native protein.

**Table 3 ijms-24-10298-t003:** The fraction of aggregated Ph*b* (γ_agg_) estimated from the amount of the protein precipitated during centrifugation after thermal or freeze–thaw-induced denaturation.

Sample	γ_agg_, %	
	Ph*b* after Heating	Ph*b* after a Freeze–Thaw Cycle
w/o additives	34	38
+500 mM Bet	9	0
+300 mM Tre	9	7
+75 mM HPCD	40	11

**Table 4 ijms-24-10298-t004:** The secondary structure of GDH and Ph*b* in the presence of HPCD.

Sample	α-Helices, %	β-Strands, %	β-Turns, %	Unordered, %
GDH				
GDH	51	14	14	21
GDH + 1 mM HPCD	51	14	15	20
GDH + 75 mM HPCD	55	13	15	17
GDH + 150 mM HPCD	57	14	14	15
GDH_fr_ + 50 mM HPCD	54	11	16	19
Ph*b*				
Ph*b*	51	11	15	23
Ph*b* + 75 mM HPCD	41	16	18	25

## Data Availability

Not applicable.

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
