# Peer review of "Effect of Chemical Chaperones on the Stability of Proteins during Heat– or Freeze–Thaw Stress"

_ijms, 2023, doi:10.3390/ijms241210298_

Round 1
Reviewer 1 Report
The manuscript entitled "Effect of chemical chaperones on the stability of proteins during heat- or freeze-thaw stress” by Borzova et al. reports an extensive investigation of the thermal stability of the GDH protein and, in less extent of the Phb protein, and its modification provided by the presence, at different concentration, of three osmolytes (coded as Tre, Bet, Sorb) and the HCPD oligosaccharide. Authors have exploited different very conventional techniques: DLS, AUC, Densitometry and viscosimetry, DSC and CD. The experimental data are analysed with standard methods. The article is quite well written and the presentation of the results as well as the discussion is extensive and appropriate. Considering the well-known importance of the protein stability, in particular in nutraceutical and pharmacological fields, the present study merits to be published in the International Journal of Molecular Sciences. However, before publication, the following minor points should be addressed by the authors.
Abstract. There are too many details on the investiated systems and, most importantly, there is not any reference regarding the techniques used in the study.
Introduction. The cited literature regarding the preferential hydration of proteins in binary solvents should also include X-ray and neutron scattering methods. See, for example, the review [10.1007/s12551-016-0193-y].
In the introduction, the biotechnological importance of the study should be better emphasized. Again, there is no any indication about the methodologies used by the authors to investigate protein stability a high and low temperature in the presence of cosolutes.
Line 112-113. It should be specified which type of light scattering was used, static or dynamic. In alternative, it should be stated that the material and methods are reported at the end of the article Why are result shown in figure 1 called examples? Are there other results?
Figure 1. How many times were DLS experiments repeated? Which is the uncertainty of the intensity? How have error bars in panels B and D been calculated? What does the small capital B on the very left of panel B represent? In the caption, the term “initial aggregation rate” to indicate the ratio nu_0,addnu_0 is not appropriate. It is the ratio between a parameter that represents the initial aggregation rate in the presence of a compound and the one in the absence of the compound. See next points.
Line 581. The term “initial aggregation rate” to indicate nu_0 seems inappropriate. Since I is measured in counts per second, nu_0 represents the counts per squared second, i.e. could be representative of the initial aggregation but not exactly an aggregation rate.
Line 165. To better evaluate the state of protein aggregation from DLS data, it is preferable to report volumetric distribution rather than the relative intensity distribution shown in Fig. 2, panels B and D.
Eq. 2 seems wrong. Usually, in equations of this type, the value [L]_0.5 is the concentration which determines the halving of the quantity on the first side. The correct equation should be nu_0,addnu_0=1/(1+([L]/[L]_0.5)^h). The exponent h is not a simple constant, but would be a parameter similar to the Hill coefficient, which is related to the cooperativity of the process. This parameter should be added in the tables and should be discussed by authors.
Author Response
Dear Reviewer,
Thank you very much for your revision concerning our paper “Effect of chemical chaperones on the stability of proteins during heat- or freeze-thaw stress”. We thank you for your valuable comments and have taken them into account in revising our article.
Reviewers' comments:
Comment 1
“Abstract. There are too many details on the investigated systems and, most importantly, there is not any reference regarding the techniques used in the study.”
Reply
The following changes were made to the text of the abstract:
The importance of studying structural stability of proteins is determined by the structure-function relationship. Protein stability is influenced by many factors among which are freeze-thaw and thermal stress. The effect of trehalose, betaine, sorbitol and 2-hydroxypropyl-β-cyclodextrin (HPCD) on the stability and aggregation of bovine liver glutamate dehydrogenase (GDH) upon heating at 50 °C or freeze-thawing was studied by dynamic light scattering, differential scanning calorimetry, analytical ultracentrifugation and circular dichroism. A freeze-thaw cycle resulted in the complete loss of the secondary and tertiary structure, and aggregation of GDH. All the cosolutes suppressed freeze-thaw-induced and heat-induced aggregation of GDH and increased the protein thermal stability. The effective concentrations of the cosolutes during freeze-thawing were lower than during heating. Sorbitol exhibited the highest anti-aggregation activity under freeze-thaw stress, whereas the most effective agents stabilizing the tertiary structure of GDH were HPCD and betaine. HPCD and trehalose were the most effective agents suppressing GDH thermal aggregation. All the chemical chaperones stabilized various soluble oligomeric forms of GDH against both types of stress. The data on GDH were compared with the effects of the same cosolutes on glycogen phosphorylase b during thermal and freeze-thaw-induced aggregation. This research can find further application in biotechnology and pharmaceutics.
Comment 2
“Introduction. The cited literature regarding the preferential hydration of proteins in binary solvents should also include X-ray and neutron scattering methods. See, for example, the review [10.1007/s12551-016-0193-y].”
Reply
The following sentence has been inserted in Introduction on page 2 at the end of the first paragraph :
“Also, the presence of cosolutes can affect the protein hydration layer, which affects protein structure, thermodynamics, stability, and activity [ PMID: 28510051].”
Comment 3
“In the introduction, the biotechnological importance of the study should be better emphasized. Again, there is no any indication about the methodologies used by the authors to investigate protein stability at high and low temperature in the presence of cosolutes.”
Reply
The beginning of Introduction has been changed to the following:
“Protein stability attracts such a close attention of scientists because of the undeniable relation between the correct functioning of proteins and their structure, and is of great interest for the biotechnology, pharmaceutical and food industries. An understanding of protein stability is essential for optimizing the expression, purification, formulation, storage and structural studies of proteins [Deller, M.C.; Kong, L.; Rupp, B. Protein Stability: a Crystallographer’s Perspective. Acta Cryst. 2016. F72, 72–95. doi: 10.1107/S2053230X15024619].”
The following information has been added at the end of Introduction on page 3:
“The goal of this work was to compare the effect of chemical chaperones on the stability of two model proteins (GDH and Phb) during freeze-thaw and thermal stresses using several methods: the dynamic light scattering (DLS), differential scanning calorimetry (DSC), analytical ultracentrifugation (AUC) and circular dichroism spectroscopy (CD).”
Comment 4
“Line 112-113. It should be specified which type of light scattering was used, static or dynamic. In alternative, it should be stated that the material and methods are reported at the end of the article. Why are results shown in figure 1 called examples? Are there other results?”
Reply
The method used has been inserted in line 112:
“The kinetic curves of GDH aggregation at 50 °C in the absence or presence of the chemical chaperones were registered by measuring the light scattering intensity (I) increase over the time (t) using DLS .”
The kinetic curves given in Fig. 1 are indeed only examples of the kinetic curves obtained in the same conditions (thermal (panel A) or freeze-thaw-induced (panel C) aggregation). These curves are given to demonstrate the raw data for different chemical chaperones. The full amount of data contains many kinetic curves obtained for each of the compounds studied at different concentrations. To represent all of them in Figures would require many identical illustrations not compliant with the article preparation guidelines and making it difficult for the reader.
Comment 5
“Figure 1. How many times were DLS experiments repeated? Which is the uncertainty of the intensity? How have error bars in panels B and D been calculated? What does the small capital B on the very left of panel B represent? In the caption, the term “initial aggregation rate” to indicate the ratio nu_0,addnu_0 is not appropriate. It is the ratio between a parameter that represents the initial aggregation rate in the presence of a compound and the one in the absence of the compound. See next points.”
Reply
1) The experiments were repeated three times. This information is indicated in Material and Methods, line 568.
2) The uncertainty of the intensities didn’t exceed 10%.
3) The error bars correspond to the SD values for the v0 value.
4) The panel B has been corrected.
5) The following changes have been made in the Figure 1 caption:
“(B,D) The dependences of v0,add / v0 on the additive concentration for aggregation of GDH or GDHfr, respectively. v0 is the parameter representing the initial aggregation rate in the absence of the additives, v0,add − in the presence of the additives.”
Comment 6
“Line 581. The term “initial aggregation rate” to indicate nu_0 seems inappropriate. Since I is measured in counts per second, nu_0 represents the counts per squared second, i.e. could be representative of the initial aggregation but not exactly an aggregation rate.”
Reply
As it follows from Eq. (1), v0 is the initial rate of the increase in I value over time, where I is used as a parameter connected with the protein particle size and representing the aggregated state of the protein. Therefore v0 depicts the change in the protein aggregation state (aggregate size and amount over time). To connect protein particle size directly with the amount of photons registered by the detector would overcomplicate the kinetic analysis, make it dependent on the equipment used and require additional photophysical interpretation of the results, while not providing any more useful information on the protein aggregation itself.
The following corrections have been made in line 582:
“...v0 is the parameter characterizing the initial aggregation rate at the stage of aggregates growth…”
and in line 589:
“...v0,add is the parameter characterizing the initial aggregation rate in the presence of the additive studied,..”
Comment 7
“Line 165. To better evaluate the state of protein aggregation from DLS data, it is preferable to report volumetric distribution rather than the relative intensity distribution shown in Fig. 2, panels B and D.”
Reply
There is a particle number distribution in Fig. 2 B and D (that is stated in the figure caption, Page 5, line 156). It is given to demonstrate the complete predominance of smaller aggregates over large particles. The latter have a higher impact on the light scattering intensity, which is depicted by the Rh value calculated from the intensity data (Fig. 2 A,C). The volumetric distribution gives more useful information when compared with mass/volume methods, such as SEC, which were not used in this work. Therefore, number distribution was demonstrated.
Comment 8
“Eq. 2 seems wrong. Usually, in equations of this type, the value [L]_0.5 is the concentration which determines the halving of the quantity on the first side. The correct equation should be nu_0,addnu_0=1/(1+([L]/[L]_0.5)^h). The exponent h is not a simple constant, but would be a parameter similar to the Hill coefficient, which is related to the cooperativity of the process. This parameter should be added in the tables and should be discussed by authors.”
Reply
- You are absolutely right, a mistake was made in typing the equation in the article. Nevertheless the correct equation was used for calculations. We have corrected this equation in the revised manuscript.
- The values of the parameter h have been added in Table 1.
The exact theoretical reasoning behind the use of Eq. (2) is given in detail in our earlier work [PMID: 27456122], which is referenced at the mention of the Eq. (2) (page XX, line xxx). In short, parameter h can be interpreted as the Hill coefficient and may represent the cooperative effects of chemical chaperone on the protein stability due to its binding at different sites of the protein molecule. However, it should be noted that chemical chaperones affect protein stability and aggregation not only by direct binding with the target protein, but also by changes in protein hydration, solution viscosity and even by the molecular crowding effect. This issue was also addressed in our previous works [PMID: 35041856], where we suggested that the anti-aggregation activity of chemical chaperones is realized by the complicated interplay of different mechanisms. Considering that, while we can use the [L]0.5 parameter for the comparative characterization of different anti-aggregation agents, the h parameter requires more careful interpretation. It can be regarded as the Hill coefficient at the relatively low concentration of cosolutes, where their effect on physico-chemical parameters is negligible. However, the higher the chemical chaperone concentration the more speculative the interpretation of h becomes.
Sincerely yours,
on behalf of the authors
Dr. Natalia A. Chebotareva, PhD, Dr.Sci
Reviewer 2 Report
The Ms ijms-2439692 presents a comparative study of the effects of a number of osmolytes and a cosolute (HPCD) on the structural integrity and thermal stability of two oligomeric proteins after freeze-thaw and heat stress. Based on DLS, AUC, DSC and CD measurements, the authors are able to identify the best anti-aggregation chemical chaperons. Results may be of interest in biotechnology and pharmaceutical industry.
In my opinion the following issues need to be addressed:
-As far as I understand, no control experiments to evaluate the actual oligomeric nature of native glutamate dehydrogenase (GDH) and glycogen phosphorylase b have been carried out. How do we know they are hexameric and dimeric proteins, respectively, within the 0,2-1 mg/mL experimental concentration range?
-In the DSC and CD experiments, it is not clear in the Method section how protein concentration is measured after the “freezing-thawing”. Are the samples first centrifuged to remove the aggregates? Errors in protein concentrations in these techniques have a direct impact on the results.
-It is stated that up to 68% of GDH form insoluble aggregates in the absence of osmolytes after 10 min at 50 °C. Similarly, it would be very informative to have an estimation of the % of GDH aggregation after freezing-thawing. According to its DSC profile on Figure 4B, no significant population remains folded afterwards.
-It is quite intriguing the small calorimetric enthalpy obtained for the thermal unfolding of GDH+150mM HPCD. Actually, it is roughly half of the mean calorimetric enthalpy for the other four experiments (see Table 2). In this regard, the authors claimed that “its transition from the native to the denatured state requires less energy” (line 255). Have they considered the possibility of a different denatured state, not fully-unfolded monomeric?? In this sense it could be revealing to compare their experimental calorimetric enthalpy value with the predicted one based on empirical relationships with DASApol and DASAapol for totally unfolded states (Robertson & Murphy, Chem. Rev. 1997).
-I would suggest to remove Scheme 1. It is highly speculative and misleading.
What is “I” and under which experimental conditions is populated? Is there any evidence supporting that “O” generates from “I” and “D” simultaneously?
Is any of the represented steps an equilibrium process? In addition, the oligomeric nature of the proteins is not taken into account.
-Fits of equation 1 to experimental data on Figures 1A and C should be shown.
- What is the rationale behind the cosolute concentrations used for the AUC experiments in Figure 3? Is there any correlation with their respective [L]0.5 in Table 1?
-Please, check the units on ellipticity in Figure7. °C is not correct.
-Line 194: 350mM Tre should be 500mM.

Author Response
Thank you very much for your revision concerning our paper “Effect of chemical chaperones on the stability of proteins during heat- or freeze-thaw stress”. We thank you for your valuable comments and have taken them into account in revising our article.
